# Using Hybrid Algorithms of Human Detection Technique for Detecting Indoor Disaster Victims

**Ho-Won Lee** [1] , **Kyong-Oh Lee** [1] , **Ji-Hye Bae** [2] , **Se-Yeob Kim** [1] **and Yoon-Young Park** [1,*]

1    Department of Computer and Electronics Convergence Engineering, Sunmoon University, Asan-si 31460, Korea
2    Department of IT Education, Sunmoon University, Asan-si 31460, Korea
*    Correspondence: yypark@sunmoon.ac.kr; Tel.: +82-10-8803-2365

**Abstract:** When an indoor disaster occurs, the disaster site can become very difficult to escape from due to the scenario or building. Most people evacuate when a disaster situation occurs, but there are also disaster victims who cannot evacuate and are isolated. Isolated disaster victims often cannot move quickly because they do not have all the necessary information about the disaster, and secondary damage can occur. Rescue workers must rescue disaster victims quickly, before secondary damage occurs, but it is not always easy to locate isolated victims within a disaster site. In addition, rescue operators can also suffer from secondary damage because they are exposed to disaster situations. We present a HHD technique that can detect isolated victims in indoor disasters relatively quickly, especially when covered by fire smoke, by merging one-stage detectors YOLO and RetinaNet. HHD is a technique with a high human detection rate compared to other techniques while using a 1-stage detector method that combines YOLO and RetinaNet. Therefore, the HHD of this paper can be beneficial in future indoor disaster situations.

**Keywords:** indoor disaster situation; detect victims; YOLO; RetinaNet; hybrid human detection; 1-stage detector; image processing; object detection

## 1. Introduction

Currently, people spend 80–90% of their time in buildings, which have become larger and more complex due to the development of architectural technology [1]. People can be exposed to disaster situations such as gas leaks and fires when they spend more time indoors. In addition, in recent years, the risk of disasters is also increasing with the development of building technologies such as residential and commercial complexes, new apartments, and government offices. There are various reasons why disasters occur, but those due to fire, in particular, are frequent.

When a disaster such as a fire occurs indoors, the "golden" time is five minutes [2]. In an indoor fire situation, smoke is more dangerous for the victims than flames; more than 60% of deaths due to fire are suffocation or death from gas and smoke [3]. When an indoor disaster occurs, most people recognise the situation and evacuate. However, often, victims cannot escape in time due to late situational awareness or for personal reasons.

Disaster victims who have yet to evacuate often do not know about the severity of the situation because it is challenging to see due to smoke caused by a fire situation. Therefore, rescue activities should be carried out as quickly as possible to prevent secondary damage. Rescue workers also have difficulty with low visibility, so they often have to carry out rescue operations using the cries of disaster victims. As a result, rescue workers are also exposed to disaster situations, which can result in injury due to secondary damage. This is because they have to enter the building interior to carry out rescue operations without knowing the location of the disaster victim.

Most previous studies on this subject guide the rescue route from the "current" location [4–6] to the escape route for those who were not able to evacuate the disaster site early

on [7–10]. Studies to detect people isolated in buildings are also being conducted; however, most of them are aimed at detecting fire disasters rather than people [11–13]. Research on detecting people using autonomous mobile robots [14–17] or thermal imaging [18,19] and infrared cameras [20] is also in progress.

Detecting people using an autonomous mobile robot or infrared camera can ensure the safety of rescuers and can detect people who are obscured by smoke, since thermal imaging can penetrate the smoke. However, having an autonomous mobile robot reside indoors or attaching an infrared camera to the building is very expensive. In addition, for an autonomous mobile robot to know the current state of the building layout, it must have previously scanned the inside of the building to create a map. Therefore, the time and cost of scanning and processing the map must be considered.

The time between the occurrence of a disaster and the arrival of rescuers to begin operations must not be longer than five minutes. This is not enough time for rescuers to arrive and locate isolated victims by deploying an autonomous mobile robot; therefore, early response is critical. The best way to identify the whereabouts of disaster victims in the early stages of a disaster is to use closed circuit television (CCTV). Cameras for this purpose are often installed in buildings, so it is possible to identify the victims' whereabouts quickly using video footage.

There are few studies on detecting people obscured by fire smoke [21,22] because it is dangerous to simulate a situation where people are at risk of fire. For this reason, the majority of the research on this subject detects disaster situations rather than people [23–30]. However, as previously explained, there are still situations where people cannot escape. Therefore, in this paper, we propose a method for detecting people according to changes in fire-smoke concentrations by overlaying images from inside the building with a smoke filter.

The method uses CCTV installed in buildings and has a faster detection speed and higher accuracy than the techniques presented in previous studies. To this end, we used You Only Look Once (YOLO) [31] and RetinaNet [32], which are both 1-stage methods (rather than 2-stage such as the convolutional neural network (CNN) or Fast-RNN [13]), and by merging them, we designed a method to detect disaster victims faster and more accurately.

The purpose of this paper is to more quickly and accurately detect disaster victims surrounded by smoke in an indoor fire disaster situation. For this purpose, we examined application of the optimal Intersection Over Union (IoU) value by merging the YOLO and RetinaNet methods.

The structure of this paper is as follows. Section 2 presents related research, and Section 3 describes the design process for the technique proposed in this study. In Section 4, the test and experimental results for the design are explained, and finally, in Section 5, conclusions are presented.

## 2. Related Works

In general, 2-stage methods use anchors to suggest objects for classification and regression [33,34], whereas 1-stage methods [30,35,36] proceed directly to classification (i.e., the anchor box is modified without object suggestion).

In this chapter, the following three studies related to human detection in indoor disaster situations are discussed: detecting people in fire smoke; detecting disaster victims using CNN; and detecting disaster victims using YOLO.

### 2.1. Human Detection in an Area with Fire Smoke

The first study proposes a novel method combining a situational awareness framework and automatic visual smoke detection [33]. The detection work was carried out by learning information about scenarios with smoke and fire, along with information about people. Seventy percent of the training dataset was trained with the k-nearest neighbour (KNN) [37] classifier.

Figure 1 shows two examples of detecting a person obscured by smoke using the KNN classifier and the system classifier [34]. An adaptive background subtraction algorithm

was used to identify moving objects based on the dynamic characteristics of smoke. Two features, colour and fuzziness, are applied to filter regions without smoke motion. Only an area that satisfies certain colour analysis and fuzzy characteristics is selected as an acting candidate area. There are still problems determining the existence of humans for various reasons, including the smoke itself. However, it is possible to determine a person's presence by detecting only a part of the body as a feature point, if the person is partially covered by smoke.

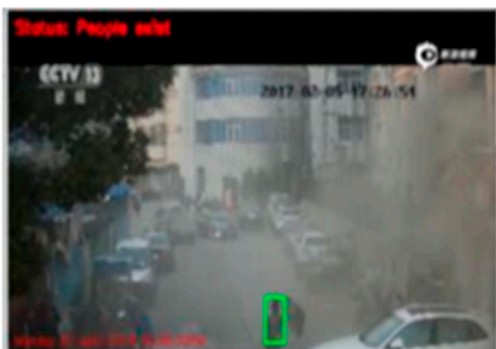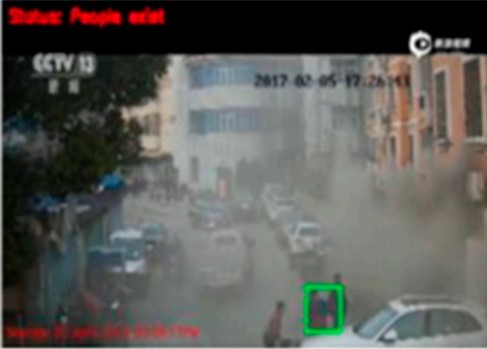

**Figure 1.** Two examples of human detection in smoke scenes, indicated with green boxes.

### 2.2. Victim Detection Using Convolutional Neural Networks

Another study describes the detection of people and pets in any location by providing an infrared (IR) image with location information during combustion to a convolutional neural network [35]. Two methods are proposed to develop a CNN model for detecting people and pets at high temperatures. The first method consists of a feed-forward design that categorises objects displayed in the IR image into three classes. The second method consists of a cascading two-step CNN design that separates the classification decisions at each step.

IR images are captured at the combustion site and transmitted to the base station via an autonomous embedded system vehicle [30]. The CNN model indicates whether a person or pet is detected in the IR image on the primary computer. Next, it analyses each IR image to determine one of three classes: "people", "pet", or "no victims". The proposed CNN model design improves the safety and performance of firefighters when evacuating victims from fires by setting priorities for rescue protocols.

However, since the above study uses IR images, the object's shape is not precise enough. Given that it is a study aimed at searching for disaster victims, the details for classifying disaster victims are insufficient.

### 2.3. Detection of Natural Disaster Victims Using YOLO

Finally, studies were conducted using the YOLO method to take images of victims using drones that help locate victims in complex or vulnerable locations, to direct human access. These use image processing to design natural disaster victim detection systems [38].

As shown in Figure 2, when an image is used as input to the network model, the output is calculated according to the parameters and structure of the model. This output includes image category information, coordinate information corresponding to the bounding box, and other corresponding information.

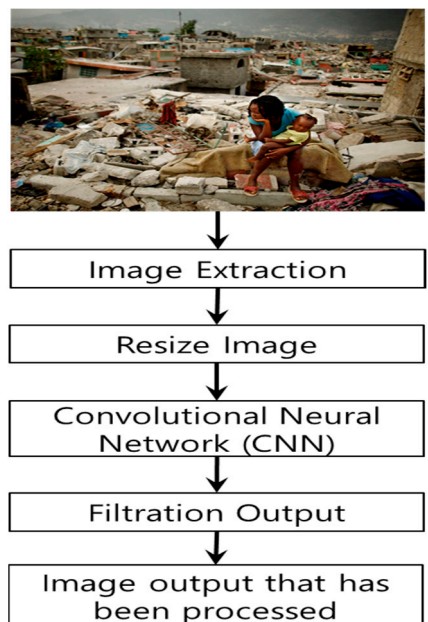

**Figure 2.** Steps in detecting disaster victims using the YOLO method.

The process to be performed includes predicting the coordinates and position of the bounding box containing the object, the probability of the bounding box containing the object, and the probability of each object of the bounding box being contained in the specified class. Next, the output image of the CNN process performs a filtering process to determine more specific objects. The output video contains information about the name of each detected object.

The data set it used contains 200 images: 100 for training and 100 for testing. The training data were trained 3000 times, and the experiment had an accuracy of 89%. However, one disadvantage is that several factors, such as the background of the object in the image, as well as the position, height, and distance, affect the detection result; this can significantly reduce the accuracy. In addition, detecting disaster victims using YOLO alone without considering the exact disaster situation has a very high probability that the detection accuracy will be low when the disaster victim is in a different situation.

## 3. Hybrid Human Detection Method

The detection method design for this study is described next. For YOLO, the type and location of an object can be guessed just by looking at the image [39]. In addition, because the background is not part of a class and only objects are designated as candidates, it is a fast and simple process with a relatively high mAP (mean average performance). However, it has low accuracy for small objects.

For RetinaNet, the background and object classes are separate, so when there are significantly more areas in the image than the area in which the object is located, the loss function is used to increase accuracy [40].

Figure 3 shows a flowchart of our proposed disaster victim detection task. False positive (FP) and false negative (FN) classification results are obtained when the first detection operation is performed using YOLOv3, and then the secondary detection operation is performed using RetinaNet. The secondary detection operation is performed after excluding the operation results, classified as true positive (TP) and true negative (TN) in the primary detection operation.

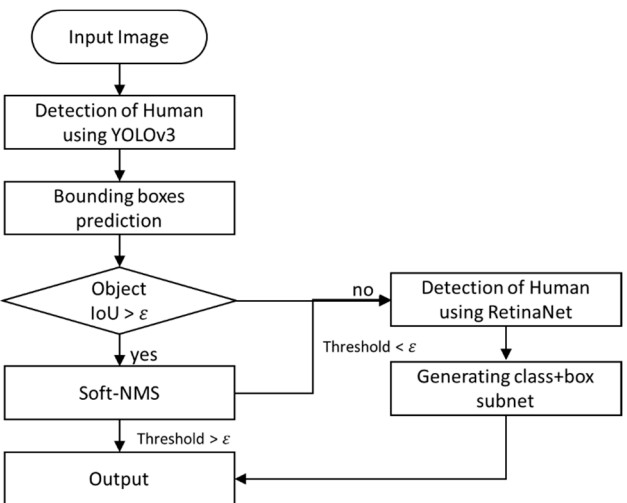

**Figure 3.** Hybrid Human Detection Flowchart.

When fire causes an indoor disaster situation, a person at the scene can be obscured by smoke. To detect a person in such a situation, an image is selected and learned through a machine learning module. The accuracy of human detection varies with smoke concentration. At this time, the optimal IoU value is found by trial and error; this is the most crucial factor in saving lives in an indoor disaster. For this reason, we designed a hybrid human detection (HHD) method that focuses on finding the optimal IoU value using both YOLOv3 and RetinaNet.

The proposed HHD task is divided into four significant steps, as shown in Figure 4. First, set the candidate group of objects in the input image. Next, the input image is divided into a grid (dimensions $S \times S$). Each grid cell predicts $B$ bounding boxes along with a confidence score. Each bounding box predicts the reliability of the x- and y-coordinates, the value h for height, and the value w for area. The confidence predicts the ground truth box and IoU of the predicted box, and an optimal IoU value is derived. One class is predicted per cell.

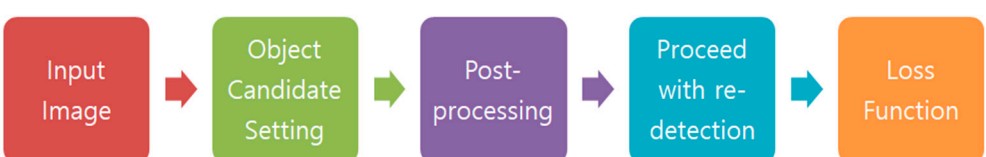

**Figure 4.** Hybrid human detection 4-step workflow.

Second, the bounding box with the highest reliability is used for an object in post-processing, and the remaining bounding boxes are removed. The bounding box with the highest confidence is found. Through this series of processes, objects classified as FN and FP are searched for as a result of classification.

Third, using RetinaNet, re-detection is performed for objects classified as FN and FP, and the anchor box is created. RetinaNet is divided into a subnet containing object and bounding box coordinate information. By separating the background and the object, the detection operation is performed more accurately by focusing on the object detection.

If proceeding at this point, much loss will occur. Because only people are detected among objects, an imbalance between foreground and background occurs. To resolve this imbalance, focal loss is applied in the last step. Focal loss applies a shallow loss value to data that are already classified. In addition, it is possible to represent the detection result more accurately because it gives more weight to the loss by concentrating on the misclassified data.

The creation of bounding boxes in the first step is shown in Figure 5. $p_h$ and $p_w$ represent the height and width of the anchor box; $t_x$, $t_y$, $t_w$, and $t_h$ represent prediction

values; and $b_x$, $b_y$, $b_w$, and $b_h$ represent post-processing information. $b$ is the predicted offset of the bounding box to be used in the anchor box. $c$ indicates the offset of the upper left end of each grid cell. The object is detected using the final value of $b$ and the ground truth IoU, calculated by the following equation:

$$\text{IoU} = \frac{area\left(B_{gt} \cap B_b\right)}{area\left(B_{gt} \cup B_b\right)}, \tag{1}$$

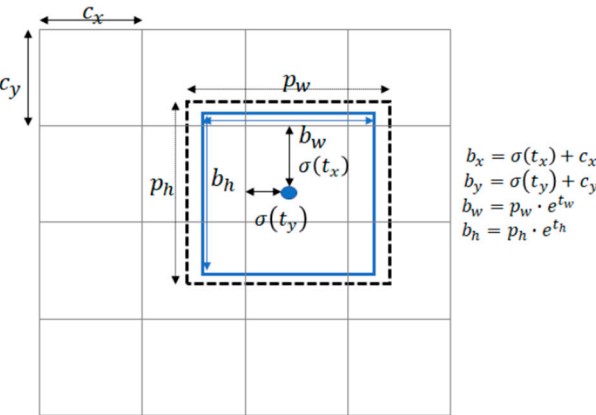

**Figure 5.** Prediction of bounding boxes.

IoU is a method for evaluating two boxes—$B_{gt}$ (ground truth), the location bounding box of an actual object, and $B_p$ (prediction), the predicted bounding box—through overlapping areas. Larger overlapping areas indicate better evaluations. The existence of an object is determined by the degree of overlap between the ground truth and bounding boxes. In previous studies, the IoU value is dynamically changed because it is obtained using a video. However, this makes it difficult to determine accurately the existence of an object because the existence of a person obscured by smoke must be explored through fragmentary images. Therefore, it is necessary to find the optimal IoU value.

In addition, our goal was to detect people obscured by smoke in an indoor disaster situation, in particular, a fire, rather than detecting a person in a generic situation. Thus, the occlusion phenomenon occurs for humans as objects being detected. In this case, a problem arises when the correct box may be deleted. Using the following formula prevents the box from being deleted:

$$S_i = \begin{cases} S_i, & IoU - R_{DIoU}(M, B_i) < \epsilon \\ 0, & IoU - R_{DIoU}(M, B_i) \geq \epsilon \end{cases} \tag{2}$$

$\epsilon$ represents the threshold and is also the classification score. If the centre distance is long and the IoU is large, it is possible to detect another object, so the issue of deleting the correct box can be prevented.

YOLO predicts multiple bounding boxes for each grid cell. To compute the loss for true positives, we need to select one box that best contains the detected objects. The formula below is for performing this process.

$$\lambda_{coord} \sum_{i=0}^{s^2} \sum_{j=0}^{B} 1_{ij}^{obj} \left[ (x_i - \hat{x}_i)^2 + (y_i - \hat{y}_i)^2 \right] + \lambda_{coord} \sum_{i=0}^{s^2} \sum_{j=0}^{B} 1_{ij}^{obj} \left[ (\sqrt{w_i} - \sqrt{\hat{w}_i})^2 + (\sqrt{h_i} - \sqrt{\hat{h}_i})^2 \right] \tag{3}$$

where $s$ is the number of grids, and $B$ is the number of bounding boxes predicted by each grid cell. A $7 \times 7$ grid predicts two bounding boxes and optimises by finding a loss only when an object is within a grid cell.

Specifically, YOLO divides the image into $7 \times 7$ grid cells and predicts two candidates for objects of various sizes centred on each grid cell. In the case of the 2-stage method, if

more than 1000 candidates are proposed, YOLO proposes only $7 \times 7 \times 2 = 98$ candidates, so the performance is worse. Furthermore, the detection accuracy is significantly lower when there are several objects surrounding one object, i.e., if there are several objects in one cell the detection accuracy decreases.

After performing the primary detection task with YOLO, the second detection task is performed using RetinaNet for objects classified as FN and FP that were not detected. It divides the missing object into the subnet of object information and the coordinates of the bounding box. It then classifies the bounding box and predicts the distance between the bounding box and the ground truth object box. RetinaNet can also collect background information and focus on the object, which increases accuracy.

If the background and foreground become unbalanced, the loss rate of detection accuracy increases, and a loss function should be applied to reduce this imbalance. The following equations are used to calculate the loss function to reduce the loss rate:

$$p_t = \begin{cases} p & if\ y = 1 \\ 1 - p & otherwise' \end{cases} \tag{4}$$

$$CE(p, y) = CE(p_t) = -\log(p_t), \tag{5}$$

$p$ is the value predicted by the model, while y is the value for the ground truth class. In Equation (4), $p$ is 1 or $-1$; it represents the ground-truth class. $p_t$ is between 0 and 1; it represents the class probability for a class predicted by the model. Equation (5) defines a function that is slightly more convenient than Equation (4). When $p_t \geq 0.5$, it is easy to classify; the easy example has a slight loss, but it takes up most of the loss when the number increases. Therefore, the following formula is used to reduce the effect of this easy example on the loss:

$$FL(p_t) = \begin{cases} -(1 - p_t)^\gamma \log(p_t), & if\ y = 1 \\ -(1 - (1 - p_t))^\gamma \log(1 - p_t), & otherwise \end{cases} \tag{6}$$

Equation (6) is an expression for focal loss. Focal loss is a loss function that down-weights cases that are easy to classify; it learns by focusing on difficult classification problems. The modulating factor $(1 - p_t)^\gamma$ and the tuneable focusing parameter $\gamma$ are added to CE. As $\gamma$ becomes larger than 0, the difference in loss values between well-detected and non-detected objects becomes more evident.

## 4. Applications and Results of the Hybrid Human Detection Method

Figure 6 shows a representative image data set in which a smoke filter is applied to an image of people indoors. The concentration of the smoke filter was adjusted for transparency, starting at 75% and increasing to 85% in 1% increments. The same image filter and transparency were applied to all test image datasets.

For smoke concentrations greater than 90%, comparisons between techniques are meaningless because the concentration is too thick to identify a person. We performed and compared detection using YOLO only, RetinaNet only, and HHD.

Figure 7 shows how to set the object candidate group during the work to detect disaster victims covered by smoke using YOLOv3. Figure 7a shows the generation of bounding boxes surrounding predicted objects. Figure 7b shows the areas where it is predicted that there are humans. Following this step, and further preventing the correct bounding box from being deleted due to the phenomenon of occlusion of a human by smoke (Equation (2)), only the most reliable bounding box is left, and the final result is shown in Figure 8 below.

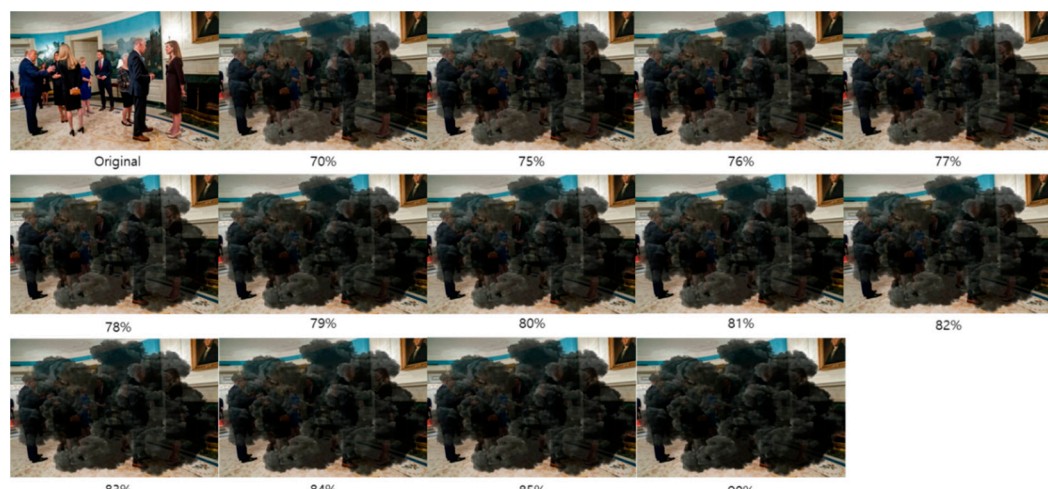

**Figure 6.** Images showing varying smoke concentration.

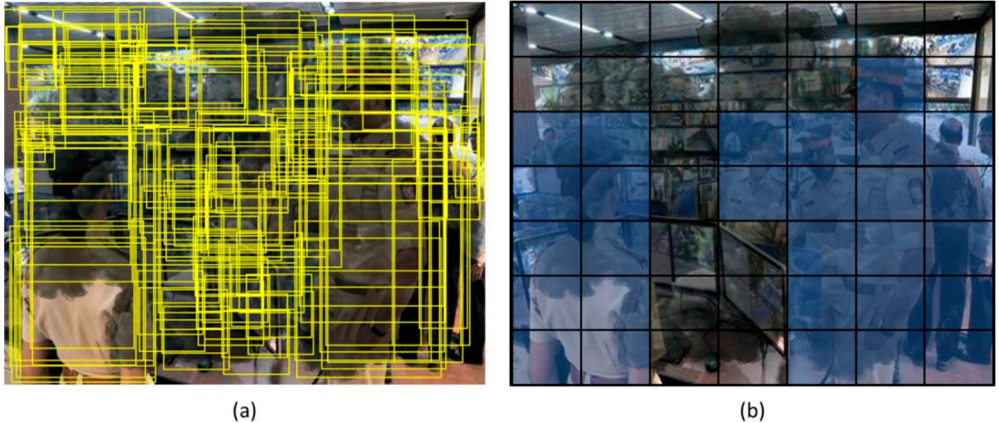

**Figure 7.** (**a**) Bounding boxes + confidence; (**b**) Class probability map.

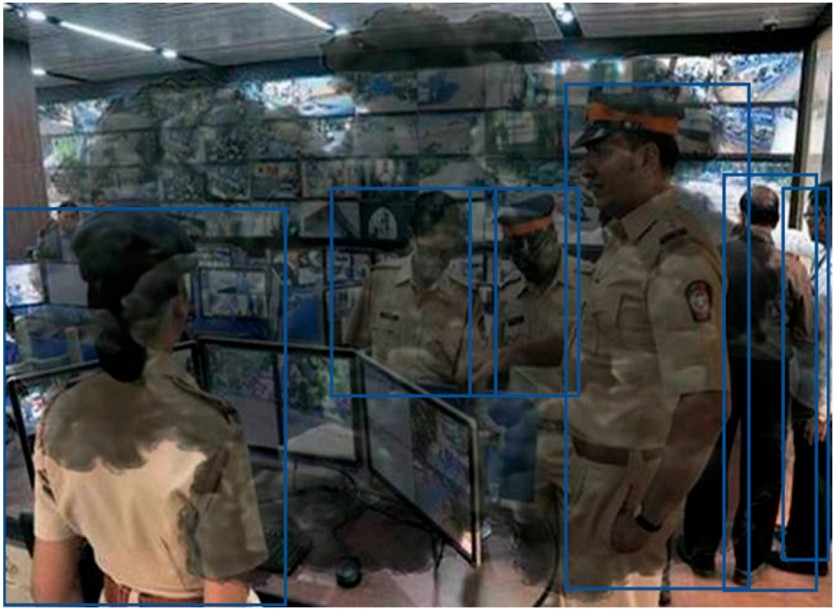

**Figure 8.** Image showing results of human detection using YOLOv3.

Although it is predicted that there is an object in the ground truth box, it results in an FN classification that cannot extract the object, as shown in Figure 9. Although there are no objects in these FN or ground truth boxes, secondary detection is performed using RetinaNet only for objects classified as FP, which predict that there are objects (Figure 10). RetinaNet puts an anchor box on each point per feature map.

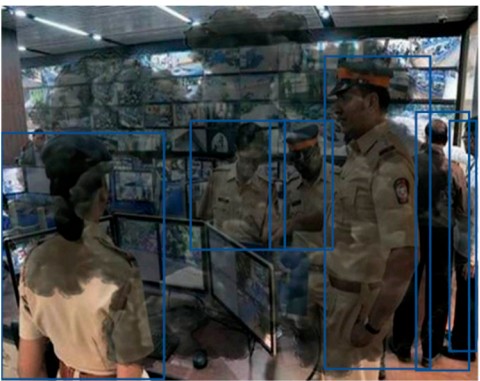 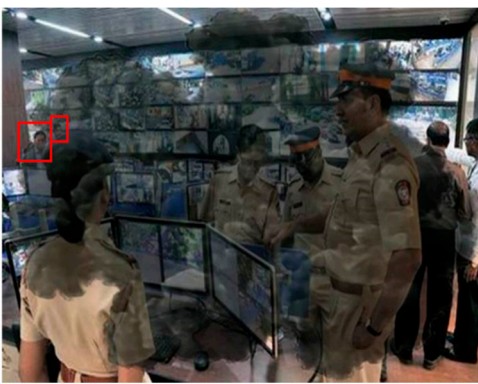

**Figure 9.** Results of classification to false negative (YOLO).

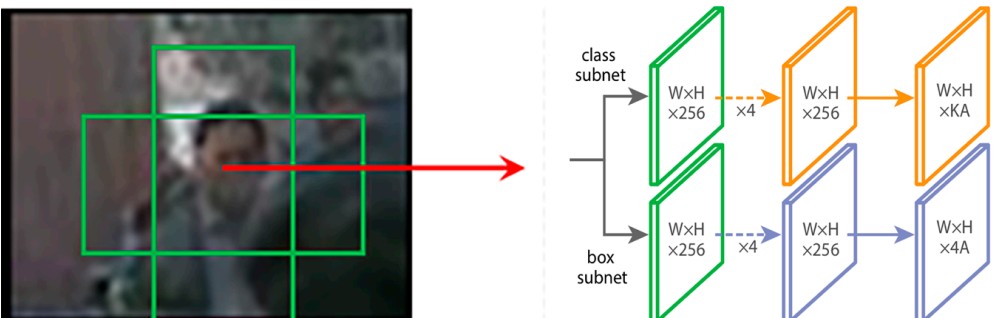

**Figure 10.** Results of classification to false negative (RetinaNet).

For objects classified as FN and FP, the subnet of object information and that of the bounding box coordinate information are divided, and then the bounding box is classified. Next, the distance between the bounding box and the ground truth box is predicted. If there are more objects in the background than in the foreground, class imbalance occurs. To prevent this phenomenon, the focal loss function can be used. Figure 11 shows the results obtained for objects classified as FN and FP using this method.

Figure 12 below shows the relative accuracies of the YOLO, RetinaNet, and HHD methods when applying IoU values of 0.3, 0.5, and 0.7. For a smoke concentration of 70%, the accuracy of the three methods was similar, but for 75% and higher, the HHD method was more accurate.

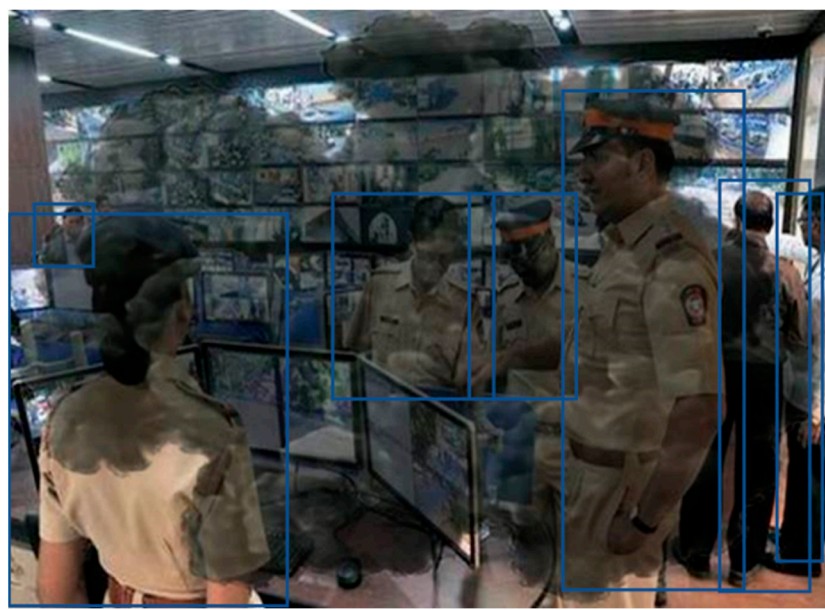

**Figure 11.** Image showing results of secondary detection using RetinaNet.

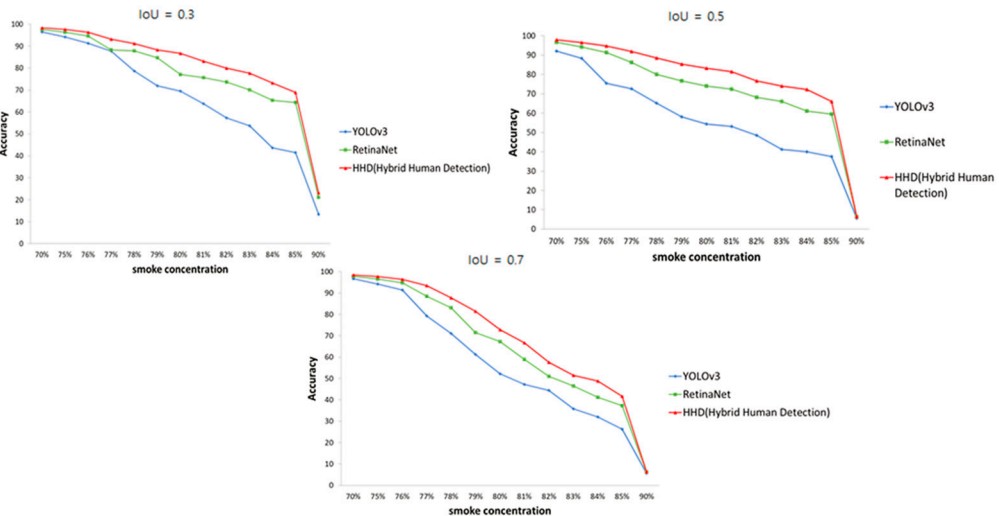

**Figure 12.** Accuracy of the YOLO, RetinaNet, and HHD methods vs. smoke concentration for IoU values of 0.3, 0.5, and 0.7.

A value of IoU = 0.3 gave the highest overall detection accuracy, however, as shown in Figure 13, a person was detected in a part of the image where there was no person. Therefore, IoU = 0.3 is not ideal for this task. IoU = 0.7 gave more accurate results than IoU = 0.5 for 70–79% smoke concentration. However, the accuracy is lower for a smoke concentration of 80% and higher. As the smoke thickened and obscured more people, the accuracy decreased.

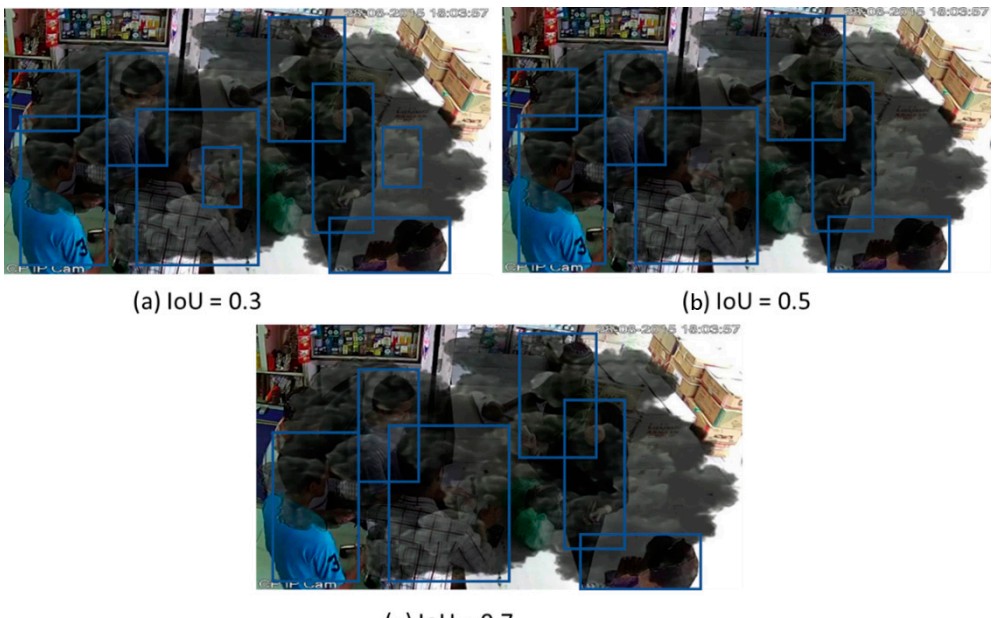

(a) IoU = 0.3    (b) IoU = 0.5

(c) IoU = 0.7

**Figure 13.** Images showing the results of detection of disaster victims obscured by smoke for IoU = (**a**) 0.3, (**b**) 0.5, and (**c**) 0.7.

Figure 13 shows images of human detection results for the three IoU values: (a) IoU = 0.3, (b) IoU = 0.5, and (c) IoU = 0.7. For IoU = 0.3, some of the smoke was recognised as a human in addition to the actual humans. For IoU = 0.7, when the smoke was thick (higher concentration), the shape of the person was not visible and could not be detected.

Even with IoU = 0.5, the detection accuracy was not high for scenarios where only a part of the human body was visible. To minimise the scenarios in which non-human objects are mistakenly recognised as human or cannot be detected when a part of the human body is covered, we found the optimum value of IoU with the highest detection accuracy for each smoke concentration. Figure 14 shows the results when IoU = 0.3, 0.5, 0.7, and optimal IoU were assigned. Table 1 gives the numerical values of the graph in Figure 14. The blue highlighted figures in the table represent the optimal IoU values for each smoke concentration.

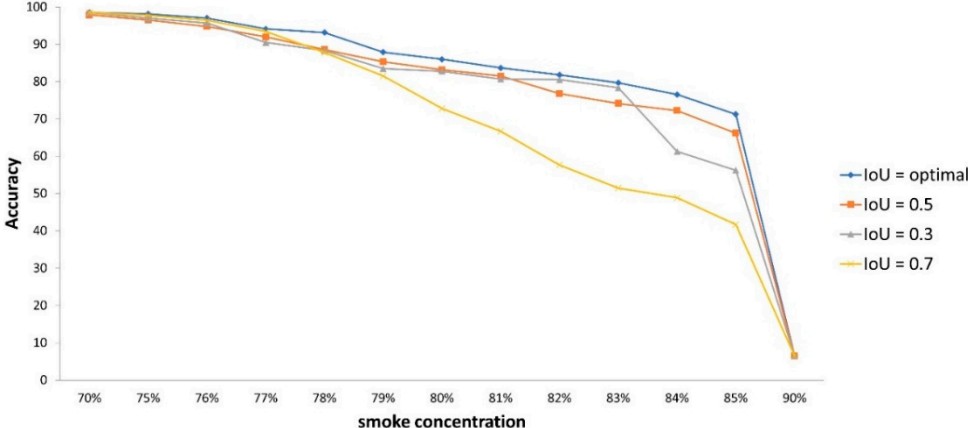

**Figure 14.** Average accuracy vs. smoke concentration for IoU = 0.3, 0.5, 0.7, and optimal IoU.

**Table 1.** IoU = 0.3, 0.5, 0.7, and average accuracy for optimal IoU values (highlighted in blue) by smoke concentration.

| | 70% | 75% | 76% | 77% | 78% | 79% | 80% | 81% | 82% | 83% | 84% | 85% | 90% |
|---|---|---|---|---|---|---|---|---|---|---|---|---|---|
| IoU = | 0.73 | 0.73 | 0.72 | 0.73 | 0.45 | 0.46 | 0.55 | 0.57 | 0.56 | 0.56 | 0.55 | 0.56 | 0.63 |
| Optimal | 98.51 | 98.14 | 97.03 | 94.10 | 93.14 | 87.88 | 95.98 | 83.65 | 81.78 | 79.65 | 75.51 | 71.24 | 6.47 |
| IoU = 0.3 | 98.33 | 97.65 | 96.31 | 93.15 | 91.18 | 88.29 | 86.67 | 83.12 | 80.01 | 77.66 | 73.24 | 68.88 | 23.11 |
| IoU = 0.5 | 97.87 | 96.52 | 94.77 | 91.88 | 88.57 | 85.31 | 83.17 | 81.48 | 76.76 | 74.11 | 72.23 | 66.15 | 6.45 |
| IoU = 0.7 | 98.48 | 97.78 | 96.45 | 93.41 | 87.77 | 81.46 | 72.78 | 66.66 | 57.65 | 51.45 | 48.87 | 41.65 | 6.45 |

In Table 1 and Figure 14, IoU = 0.3 showed higher average accuracy than the optimal IoU for smoke concentrations of 79%, 80%, and 90%; objects covered by the smoke were regarded as persons, as shown in Figure 13a above.

Table 2 shows the average detection rates of the YOLO, RetinaNet, and HHD methods for different smoke concentrations. On average, YOLO was the fastest with an average speed of 1 s, and RetinaNet had an average speed of 2–3 s. HHD was slightly slower than YOLO but faster than RetinaNet by 1 s, on average.

**Table 2.** Average detection speeds for YOLO, RetinaNet, and hybrid human detection for varying smoke concentrations.

| | 70% | 75% | 80% | 85% | 90% |
|---|---|---|---|---|---|
| YOLOv3 | 1.7 s | 1.3 s | 1.1 s | 1.2 s | 0.9 s |
| RetinaNet | 3.1 s | 2.6 s | 2.7 s | 3.2 s | 2.6 s |
| HHD | 2.6 s | 1.5 s | 1.8 s | 2.1 s | 1.2 s |

Figure 15 shows the precision and recall by smoke concentration when IoU values of 0.3, 0.5, 0.7, and optimal values are applied to the HHD method. Precision is defined by the percentage of correct detections among all detection results, and recall is defined by the percentage of correctly detected objects in the ground truth box. For the recall, the reason the value exceeds 1 when the value of IoU is 0.3 is that objects were incorrectly recognised as human.

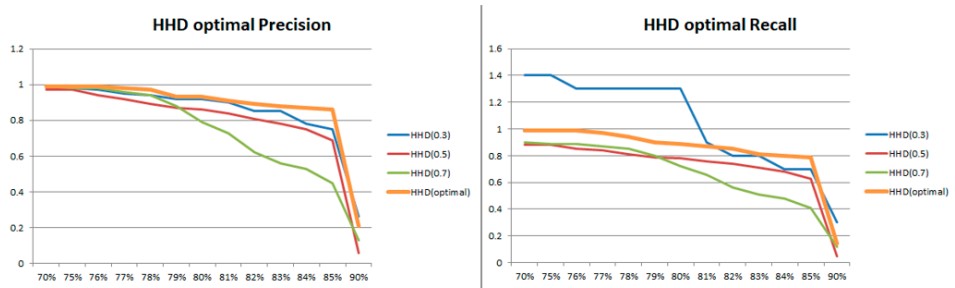

**Figure 15.** Precision and recall graph, by smoke concentration, with IoU = 0.3, 0.5, 0.7, and IoU optimal values, for the HHD method.

Since a smoke filter was applied to an image rather than a video, it was possible to derive an optimal IoU value for each smoke concentration. When IoU = 0.3, as mentioned above, a non-human object was recognised as a human. Furthermore, compared with IoU = 0.5 and 0.7, the method in which the optimal value of IoU suggested here was found and applied was higher than 1% and as high as 6%.

These results mean that victims of smoky disasters can be detected more accurately, and people in indoor disasters can be detected more accurately in urgent situations. This information can be processed and delivered to rescuers to minimise victims of indoor disasters.

## 5. Discussion

In a previous study of human detection in smoke, the classification of persons in smoke was biased because it included smoke in the detection criteria [33]. However, in the present study, even when a person was covered by smoke, it was possible to detect the person by detecting a part of that person's body as a feature point. In another study, IR cameras were used for detection; class imbalances due to a large number of human data sets were balanced by oversampling [35]. In the present study, we avoided class imbalance by using the focal loss method. We also developed a hybrid human detection method that merged the learning methods of YOLO and RetinaNet; we improved the accuracy of the new method by reducing loss to increase the detection rate and by developing an approach to determine an optimal IoU value through the dynamic assignment of multiple IoU values.

However, because we applied the smoke filter to images rather than videos, our approach determines the IoU value separately for each smoke concentration. Therefore, when our approach is applied to videos, different optimal IoU values might be obtained. In addition, because we tested our approach on random non-disaster situations, rather than real disaster situations, the applicability of our approach to real disaster situations is limited; it requires further training and optimisation.

## 6. Summary and Conclusions

In this paper, we propose a method to more accurately detect disaster victims who have not yet evacuated from an indoor disaster situation, especially those isolated due to fire smoke. The work was carried out using the 1-stage detector method, and a HHD method combining the YOLO and RetinaNet methods was proposed. Since the image of a part of a person's body is learned, it could be detected accurately even if the entire shape of a person was not visible, and we also increased the detection accuracy by using CCTV images. In addition, the work was carried out to detect disaster victims by applying a unique environment, called a disaster.

Because a person's body can be partially or completely obscured by smoke, the accuracy is significantly lower than for detection in a non-smoky situation. In this study, detecting disaster victims hidden by smoke is carried out using the proposed HHD method. HHD uses the YOLO and RetinaNet methods together, improves detection accuracy by repeating the search for FN and FP classifications, and also finds the optimal IoU value to produce better results.

For YOLO, high detection accuracy was found when the human body was completely visible. However, when the smoke thickened and only a part of the human body was visible, or the human body was blurred, the accuracy was low. In this case, detection may not be possible due to the overlap of objects or class imbalance. For RetinaNet, since there are classes divided by background and objects, the detection accuracy was higher than that of YOLO, but there was a significant difference with YOLO in terms of speed. Furthermore, some objects were missed completely.

When HHD was used, the detection accuracy was higher than that of YOLO. When compared with RetinaNet, the detection accuracy was not significantly different from that of YOLO, but the result could be arrived at more quickly. When comparing the detection accuracies of YOLO, RetinaNet, and HHD, on average, when HHD and YOLO were applied with IoU = 0.5, the most significant deviation was shown, and the detection accuracy ranged from 3% to 20%. When comparing HHD and RetinaNet, the most significant deviation was found when IoU = 0.7 was applied, and the high detection accuracy ranged from 1% to 9%. The parameters showing the most significant deviation on average was for the difference in accuracy starting with a smoke concentration of 80%. Furthermore, the differences between using values of IoU = 0.3, 0.5, and 0.7 were calculated for HHD and the optimal IoU value was found and applied.

Our results show that the HHD method proposed in this paper produces better results than when YOLO or RetinaNet is used alone. Utilising videos and partial smoke coverage of people might further improve the accuracy of the HDD method but requires further

testing. Our method could constitute an essential factor in identifying victims in indoor disaster situations, especially where there is fire and smoke. This information has the advantage of contributing to the prevention of additional disaster victims because rescue work could be conducted more quickly.

**Author Contributions:** Conceptualization, S.-Y.K., K.-O.L. and J.-H.B.; formal analysis, S.-Y.K., K.-O.L. and Y.-Y.P.; investigation, S.-Y.K., K.-O.L. and Y.-Y.P.; project administration, S.-Y.K. and H.-W.L.; resources, J.-H.B. and H.-W.L.; software, S.-Y.K.; supervision, K.-O.L., J.-H.B. and Y.-Y.P.; validation, S.-Y.K., K.-O.L., H.-W.L. and Y.-Y.P.; writing—original draft, S.-Y.K.; writing—review and editing, S.-Y.K., K.-O.L. and H.-W.L. All authors have read and agreed to the published version of the manuscript.

**Funding:** This work was supported by National Research Foundation of Korea Grant Funded by the Korean Government (NRF2021R1A2C1004651).

**Conflicts of Interest:** The authors declare no conflict of interest.

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
