# Peer review of "Using Hybrid Algorithms of Human Detection Technique for Detecting Indoor Disaster Victims"

_computation, doi:10.3390/computation10110197_

Round 1

Reviewer 1 Report

Please read the attached file. Thank you.

Author Response

Hello, this is Howon Lee of Sunmoon University.

First, thank you for reviewing our paper!

Regarding the English grammar, we would like to inform you that the initial submission of this paper has been requested to a specialized institution for verification of English grammar.
You can check the detailed certification below:
http://textcheck.com/certificate/index/LXLnRc

The answers to the comments are as follows:

  - Figure 15: What is the unit of the vertical axis? And the two figures in Figure 15 should use the same scale for easy comparison.
  -> The vertical axis of the left side graph represents the average precision value for the IoU values of the HHD, and the right side graph's vertical axis represents the HHD's average recall value.

  - How many images have you conducted in your experiment? Please discuss more about the size of the population and its characteristics for the survey.
  - How did the authors evaluate the validity of their results?
  -> This paper has classified human data sets to verify victim detection results. Of the total 66,800 pages, the learning dataset consisted of approximately 53,000 pages, the validation dataset was approximately 6600 pages, and the test image was approximately 6600 pages, resulting in a ratio of 8:1:1.

  - Have you considered the time and the speed for detecting victims? Do you think there is a tradeoff between accuracy and the speed of detecting victims? And if yes, how could you balance those viewpoints?
 -> Time and speed are key factor to detect victims in special situations such as indoor disasters. However, this study focused on detection accuracy rather than speed.

  - What are the main limitations of this approach? Please explain more about the applications of this study.
  -> The limitation of HHD presented in this study is that unexpected situation might occur with the actual fire smokes that are different from our test images because it was impossible to simulate fire hazard situations due to national laws. Currently, we are using HHD for another project that uses thermal imaging and infrared cameras to explore isolated disaster victims by robots around the disaster site when an indoor disaster situation occurs.

I am attaching a revised version of what you pointed out through comments.

Thank you!

Best regards,
Howon Lee

Reviewer 2 Report

The manuscript describes an important problem that is rarely addressed. I am not a specialist of this type of method and my comments remain rather general than precise about the computation method. The manuscript can be published after minor corrections.

Author Response

Hello, this is Howon Lee of Sunmoon University.

First, thank you for reviewing our paper!

Regarding the English grammar, we would like to inform you that the initial submission of this paper has been requested to a specialized institution for verification of English grammar.
You can check the detailed certification below:
http://textcheck.com/certificate/index/LXLnRc

The answers to the comments are as follows:

  - For acronyms, the description of the four acronyms below, which were not defined in this paper, was given at the time of first mentioning acronyms.
  CNN : Convolutional Neural Network
  YOLO : You Only Look Once
  mAP : mean Average performance
  IoU : Intersection Over Union

  - I am surprised by the relatively low number of images (100 for training and 100 for testing) especially as the training data were trained 3000 times.
  -> This is the quote from the references, and this study used a dataset of 66,800 pages. But it seems I mistyped "It" to "I".

  - I wonder how the size of the bounding boxes is determined and how it influences the results. It is likely to be dependent on the size of the persons in the image
  -> A size of the bounding box is variably determined within the algorithm as it runs.

  - What happens if the smoke cloud covers totally the image instead of this partial obscuration? It does not mean that the smoke concentration is larger than 90% but that the smoke begins to to obscure the whole scene. From the discussions in the following, it seems that the method is mostly applicable at the beginning of the fire.
  -> If a fire breaks out in reality, the smoke will spread throughout the site, unlike the image. This paper aims to improve the detection accuracy depending on how many silhouettes of a person covered by smoke are visible and how to improve this detection accuracy.

  - The difference between IoU=0.5 and the optimal IoU is very small.
  -> The values of IoU in the graph of Figure 14 on page 11 are the IoU values when the hybrid human detection (HHD) method proposed in this paper is applied. In other words, the values in the graph in Figure 14 represent the IoU values of the HHD method. The optimal IoU values in this paper are those found when the highest accuracy is obtained while adjusting these values.

I am attaching a revised version of what you pointed out through comments.

Thank you!

Best regards,
Howon Lee
